# Risk Factors for Intimate Partner Femicide–Suicide in Italy: An Ecological Approach

**DOI:** 10.3390/ijerph191610431

**Published:** 2022-08-22

**Authors:** Anna Sorrentino, Vincenza Cinquegrana, Chiara Guida

**Affiliations:** Department of Psychology, University of Campania “Luigi Vanvitelli”, 81100 Caserta, Italy

**Keywords:** intimate partner violence, femicide, intimate partner femicide–suicide, ecological approach, perpetrator risk factors

## Abstract

The present study investigated the intimate partner femicide (IPF) and intimate partner femicide–suicide (IPFS) perpetrators’ individual, relational, and contextual characteristics by analyzing, within the ecological approach, femicide cases that occurred in Italy from 2010 to 2019. On the topic, to date, scant studies examined possible differences between IPF and IPFS risk factors, and no studies have analyzed these factors by adopting an ecological systems model perspective. To this aim, archival research was carried out. Of a total of 1.207 femicides, 409 were IPF, and 227 were IPFS. Perpetrators’ age, level of employment, law enforcement membership, mental and/or physical illnesses, use of psychoactive substances, previous crimes, previous violent relationships, presence of children, previous violence in the couple, inability to accept the end of the relationship, quarrels and conflict, jealousy and the psychophysical illnesses of both authors and victims, as well as the use of firearms and victim’s request for help were analyzed. The results underlined the existence of different risk factors contributing to the IPF perpetrators’ decision to commit suicide such as perpetrators’ age, law enforcement membership, and firearm availability. These findings stress the need for specific risk assessment and management strategies for IPFS perpetrators.

## 1. Introduction

Considered the misogynous killing of women by men, motivated by hatred, contempt, pleasure, or a sense of ownership of women, femicide is the most extreme form of gender-based violence leading premature death of women [1] and is internationally recognized as “the killing of women and girls because of their gender” [2] (p. 2).

Intimate partner femicide (IPF) is a pervasive form of violence against women and represents a serious social, criminal justice, and public health issue. It could be defined as “the killings of a woman by her intimate male partner, including current or former legal spouses, common-law partners, or boyfriends” [3] (p. 78). From a gender-sensitive perspective [3,4,5], the adoption of the term IPF highlights the strong social and cultural roots underlying this form of murder. According to the United Nations Office on Drugs and Crime (UNODC) [2], in 2017, about 30,000 women had been intentionally killed by their intimate partner (34.0% of all women and girls intentionally killed worldwide).

Previous empirical results highlighted that femicide committed by an intimate current or ex-partner in some cases, is followed by the perpetrator’s suicide [3,4,5,6,7,8,9,10], a phenomenon known as intimate partner femicide–suicide (IPFS) [3,4,6,9,11,12]. Data on IPFS prevalence vary significantly. Existing studies suggest that between 18.0% and 40.0% of perpetrators of intimate partner femicide (IPF) commit suicide afterward [4,9,12,13,14,15]. Concerning the period occurring between the woman’s murder and the perpetrator’s suicide, this is not consistently described in the literature and, it can happen either immediately [4,6] or subsequently, e.g., within a week [9] or up to 60 days later [4]. Although IPFS has a lower prevalence rate [4,9,12,13,14,15] compared to the total of femicide [2] (i.e., in which no subsequent suicide occurs), they still happen with significant frequency and lead to devasting effects on families and communities [6,15].

Unfortunately, while in recent years data on IPF have increased and have been widely acknowledged [2,16,17], data on IPFS are still scant, and overall, poorly systematized [10]. In the past 30 years, most studies have focused on IPF victims [18,19] and perpetrators’ risk factors [20]. On the contrary, only a few studies specifically focused on the IPFS perpetrator’s characteristics [4,6,9,12,21]. Furthermore, almost all studies denied the phenomenon’s strong gendered nature [22] and have investigated homicide–suicide perpetrators’ characteristics regardless of the intimate partner’s gender [10,23,24,25,26,27,28,29], or have investigated IPFS within other types of lethal violence followed by suicide [30,31,32].

Previous studies on IPFS perpetrators’ risk factors showed that it is almost always characterized by an intimate male perpetrator and a female victim [3,4,5,6,9,10,15,30,33,34,35,36,37]. Perpetrators are, in most cases, of a white racial background [9,30] with a medium or low socioeconomic background [4,10], and older than the victim [4,5,12,37]. They generally used a gun for committing IPFS [3,4,5,6,9,10,11,35,38,39,40] and most often in a relationship with the current or ex-intimate partner (e.g., married, cohabiting, recently separated) [3,4,5,6,9,10,12,34,35,36,37,38,39]. Most of the existing studies were descriptive and as far as we know, none of them compared IPF and IPFS perpetrators’ individual, relational and contextual characteristics.

At the individual level, existing studies focused on gender, age, race, alcohol/drug use, mental illness, and previous crimes committed by the author [4,5,6,9,10,12,15,30,36,37,41]. Being male, white, and older are key risk factors in IPFS compared with only IPF cases [3,4,5,10,11,12,15,35,37,42,43,44]. For what concerns employment status, some studies underlined that IPFS perpetrators were of low/medium employment status, unemployed, or retired [12,21,45], or of a lower socioeconomic background (e.g., peasant farmers, unskilled artisans, low-level security personnel, or unemployed) [4]. Only two studies focused on law enforcement membership as a potential individual risk factor for IPFS [46,47]. One of these [46] analyzed 43 cases of homicide–suicide perpetrated by law enforcement officers, revealing, respectively, that in 50.0% and 44.0% of the cases victims were ex-partners and suffered intimate partner violence (IPV). Police officers seem to present a considerable number of job-related risk factors which may contribute to an increased risk of generalized aggressive response, such as “aggression, domestic violence, violence exposure, the availability of lethal weaponry, and work-related attributes” [47] (p. 97).

Regarding IPFS perpetrators’ alcohol and drug use, instead, there are contrasting results [25,30,36]. According to a case review [36], in 34.0% of homicide–suicide cases, perpetrators had detectable blood alcohol content and other substances were identifiable in 18.0% of the same group. However, according to other authors, alcohol or drug use was a risk factor in only IPF cases [20], in which perpetrators had both low levels of alcohol and drugs consumption [6,11,12,13,44,48,49,50], thus decreasing the likelihood of these perpetrators to commit suicide [5,30]. In addition, IPF perpetrators also showed a low incidence of prior involvement in criminal behaviors [6,13,41,48,51].

Contrasting results also emerged regarding the presence or absence of the perpetrators’ illness, only sometimes reported in research [3,5,6,11,12,21,41]. Some studies found that the perpetrators before the homicide–suicide showed a deterioration of health conditions, highlighted by previous contact with mental health professionals, with a higher prevalence of depression [3,5,11,12,21,37,41,42]. Conversely, other researchers have suggested that it was not a significant risk factor for IPFS because few perpetrators had a mental disorder diagnosed [52]. In addition, also the victim’s illness was found as a precipitating factor in cases of homicide followed by suicide [43,53]. Past research adopted the term “mercy killings” to describe this specific type of IPF that occurs in a consensual suicide agreement or in a hopeless situation where the perpetrator, the victim, or both, have deteriorated physical or mental health [39,54].

At the relational level, one of the most important predictors of IPFS was prior episodes of IPV: 72.0% were preceded by previous forced sexual intercourse, stalking, strangulation, and abuse during pregnancy [12]. Despite this, some authors [3,6,28] agreed suicidal killers were less likely to have a prior record for minor or violent crimes or a previous violent history against their victims compared to no suicidal killers. Furthermore, studies on IPFS confirmed the role of some precipitating factors, such as the termination of the relationship [9], sexual jealousy [37], the presence of children [12,28], the age gap between the perpetrator and the victim [4,5,37] and previous episodes of IPV [12,35]. Several studies found that IPFS occurred in marriage/cohabiting or recently separated relationships [3,4,5,6,9,10,11,12,34,37,38,39]. Therefore, the main trigger of the IPFS appeared to be jealousy, obsession, the insecurity of the perpetrators, unable to accept the divorce, estrangement, or threat of separation initiated by the female partner [3,4,6,9,10,12,28,35,37]. Archival data research [37] examined 27 cases of IPFS that occurred in Jamaica from 2007 to 2017. Again, common triggers were the offender’s obsession, sexual jealousy, and fear of separation. In addition, another study [9] in South Africa observed that IPFS victims were more likely to have ended or initiated the relationship termination than only IPF cases. According to another study [3] that compared cases of IPF and IPFS that occurred in Canada (n = 700) perpetrators who were motivated by jealousy and by sickness were almost twice as likely to commit IPFS.

Regarding age, a study carried out in Ghana on 35 cases of IPFS [4], found that in the totality of them, the male perpetrator was older than the victim (on average male offenders were 6.7 years older than their victims. Similar results were found in further studies [6,12], indicating that the increase in the perpetrators’ age was a significant risk factor for IPFS.

Concerning the presence of children, having one or more children in the household, who were not fathered by the perpetrator, was more common in IPFS cases [12], while a more recent study did not find any significant difference regarding the presence of children (both in common or not) between IPF and IPFS cases [28].

At the contextual level, the most considered factor in IPFS was firearm availability from the perpetrator, ranging across different studies from 47.0% to 92.0% [3,4,5,6,11,35,38,39,40,44,55] and used mainly in the United States and more frequently among Caucasian perpetrators; differently, cutting instrument injuries were more commonly detected in African American, Hispanic, and Asian perpetrators [56]. Concerning the availability to access a firearm, in the last year, some authors evidenced how it was used, particularly by armed security personnel, to murder a spouse/partner [10,37], prompting researchers to suggest policies that focus on effective firearm regulations to address homicide–suicide. As an additional contextual risk factor, a relevant factor in assessing the risk of escalation of IPV is the help requested by the female victim before her murder [57,58]. Unfortunately, only one study [32] investigated this contextual risk factor for IPFS revealing that in 14 IPFS cases examined, only two women contacted law enforcement, a physician, or another third party to report their previous history of IPV.

### Purpose of the Current Study

Considering the devastating effects of these phenomena [15], to better understand the precipitating risks for IPF and IPFS perpetrators and at the same time to overcome one of the greatest limitations of this topic, that is the descriptive nature of the few existing studies [4,6,32,36,37,54], the purpose of the study is to examine, by adopting the ecological approach [59,60,61], the possible role that individual, relational, and contextual risk factors have in the decision of some perpetrators to commit suicide after IPF.

The ecological approach is the most widely used model for understanding IPV and the factors associated with variations in its prevalence, e.g., [1,62,63], as it enables us to explore how the different individual, relational, and contextual risk factors operate at their respective ecological level and interact and influence each other in explaining why some IPF perpetrators committed homicide.

In particular, we try to answer the following research questions:(1)Do male IPFS perpetrators, compared with only IPF perpetrators, differ from each other in terms of individual, relational, and contextual risk factors?(2)Are some individual, relational, and contextual risk factors predictive of the perpetrators’ suicide after IPF?

## 2. Methods and Procedure

The current study was part of broader research on femicides called FATHERS (Female Abuse, Threats and Homicide: Emotional Responses and Screening), approved by the Departmental Ethics Committee (protocol nr. 18/2016).

For the study, we conducted archival research by consulting online newspapers and collecting all femicide cases in Italy, from January 2010 to December 2019 following the Vienna Declaration definition [64]. The analysis included all cases of women killings by a male intimate partner (IPF) and all cases of women killings by a male intimate partner followed by the perpetrator’s suicide (IPFS), including current or former legal spouses, common-law partners, or boyfriends [3,4].

We considered IPFS in each case of IPF followed by the perpetrators’ suicide regardless of the amount of time elapsed between IPF and the suicide. In line with other studies and methodologies, information about IPF and IPFS cases and their characteristics were investigated by consulting databases of local, regional, and national online media and newspapers; those were the most suitable sources for collecting our data of interest and were able to provide us with as much information as we needed on these phenomena [4,6,65], especially in countries, such as Italy, where no available official data, reports, and statistics exist with regard to the dimensions and risk factors analyzed in the current study.

During data collection, every effort was made to record additional information about victims and perpetrators’ individual, relational, and contextual risk factors, the homicide and/or the subsequent suicide, and consulting NGO reports and the 27ora website (the 27esimaora website is an online database of Corriere della Sera, which is one of the most spread and reputable newspaper in Italy. The 27esimaora website collects and describes stories of women victims of femicide)**.** A total of three thousand ninety-two online articles were conferred using keywords (e.g., female homicide, femicide, intimate partner femicide, woman killed, wife homicide, girlfriend homicide, a woman murdered, domestic homicide). Furthermore, the authors monitored each IPF/IPFS case over time to detect any possible missing and additional information. From 1.207 femicides in Italy in the last decade, 636 were included and considered for further analyses as identified as IPF or IPFS cases.

Data have been collected and codified according to a Fatality Review Protocol [17], developed according to the Cost Action on Femicide Guidelines (The European Union program entitled “European Cooperation in Science and Technology” launched the “Femicide across Europe” COST Action in 2013, establishing European transnational cooperation among experts addressing about femicide) [66], which included six macro thematic Areas, whose dimensions were described and reported below:Geographical data (region, province, municipality).Characteristics of the victims (name and surname, age, nationality, occupation level).Characteristics of the author (name and surname, gender, age, nationality, occupation level, law enforcement membership, previous crimes, alcohol/drug use, suicide, previous violent relationships).Characteristics related to the author/victim relationship (the type of relationship, presence/absence of children, sons/daughters in common/not in common, previous violence in the couple, and previous help requests by the victim).Criminological characteristics (the type of crime, type of weapon, motives, and place of murder).For the present study, the following set of variables was analyzed:The perpetrators’ individual-level risk factors included socio-demographic information such as gender, age, level of employment (unemployed/retired, low and high employment level), law enforcement membership, mental and physical illness, alcohol/drug use, previous crimes, and violent relationships.Among the relational risk factors that concern both the perpetrators and the victims were included: perpetrator-victim age gap, presence/absence of sons and/or daughters both in common and not in common, previous violence in the couple, inability to accept the end of the relationship, quarrels and conflict, jealousy and mental and/or physical illness both of perpetrators and victims.Finally, the contextual-level risk factors included the previous request for help from the victims and the use of a firearm as a weapon to commit IPF/IPFS.

### Data Analyses

The data collected within the database were analyzed using the SPSS statistical package (version 21.0, IBM Milano, Milan, Italy). Descriptive statistics were carried out to assess IPF/IPFS prevalence and characteristics. Then, we compared individual, relational, and contextual risk factors to investigate their possible role in some perpetrators’ decision to commit suicide after IPF by using odds ratios (ORs) because they are not influenced by sample size (unlike chi-squared, for example) [67,68]. Furthermore, to test according to the ecological system theory [59,60,61] a model capable of predicting which of the considered risk factors are associated with IPF perpetrators’ suicide, we used the hierarchical logistic regression analysis. We assessed statistical significance at least at a 0.05 level for all the statistical analyses performed. Furthermore, the individual regression coefficients’ nominal significance level (*p* < 0.05) was corrected according to the Bonferroni procedure in Step 1, Step 2, and Step 3.

## 3. Results

### 3.1. Sample

Of a total of 1.207 femicides that occurred in Italy in the last decade, 707 were IPF. A total of 636 cases were included in our sample, as no information about the perpetrators’ suicide intentions was found for the remaining 71 cases (see Table 1).

The perpetrator’s suicide followed 35.7% of IPF cases. In 88.5% of cases, perpetrators committed suicide within 24 h after the IPF. In contrast, in the remaining part of IPFS, perpetrators committed suicide in a time-lapse ranging from one day to seven months after IPF.

Perpetrators’ age ranged from 18 to 97 years (M = 51.9, SD = 17.7), while victims were aged between 16 and 93 years (M = 47.7, SD = 17.7), 75.8% were older than their victims. On average male offenders were 7.04 (SD = 5.49) years older than their victims.

Concerning IPF victims’ nationality, 77.4% were Italian. Regarding their occupational level, 27.2% resulted unemployed or retired, 27.0% had a low specialization level (e.g., hairdresser, waitress, care maid, or employee), and 4.2% had a high specialization level. A total of 9.3% of victims of IPF were affected by mental and/or physical illness.

A total of 35.2% of perpetrators were unemployed or retired, 50.0% had low specialization occupations (e.g., workers in companies, plumbers, electricians, farmers, or crafts), and 9.7% were law enforcement officers. A total of 81.4% of perpetrators were Italian, 17.6% committed previous crimes, 12.6% used psychotropic substances such as alcohol, cocaine, or marijuana, and 8.8% of perpetrators were affected by a mental and/or physical illness. In 1.6% of IPF/IPFS cases, both the perpetrator and the victim suffered from a mental and/or physical illness.

Regarding the characteristics of the relationship, in 84.3% of IPF cases, the perpetrator was the victim’s husband or ex-husband; in 57.7% of IPF cases, the perpetrator had a child in common with the victim, while in 21.5% of the cases, the children were only the victims. The most common motives for IPF were the perpetrator’s inability to accept the end of the relationship (35.7%) and jealousy (19.5%).

A total of 38.5% of perpetrators had already been violent against the victim, and 29.6% used a firearm to commit IPF.

### 3.2. Prevalence of the Individual, Relational, and Contextual Risk Factors between IPF and IPFS Perpetrators

We examined the possible existence of significant differences in terms of individual, relational, and contextual risk factors across IPF and IPFS perpetrators (see Table 2).

At the individual level, the results showed that, respectively, 43.3% of low-specialized workers, 39.0% of highly specialized professionals, and 16.7% of law enforcement committed suicide after IPF. Only being part of law enforcement was a significant predictive factor for suicide after IPF [OR = 3.23, 95% CI (1.88–5.34), *p* < 0.001]. Respectively, 9.7% and 6.2% of suicide perpetrators used substances and committed previous crimes. However, perpetrators’ prior use of substances and/or involvement in criminal behaviors seem to be protective factors for suicide. Additionally, the history of violent relationships was a predictive risk factor only for IPF.

At the relational level, 64.8% of perpetrators with sons/daughters in common with the victim committed suicide after killing the women. To have sons/daughters in common was found to be a significant risk factor for perpetrators’ suicide after homicide [OR = 1.58, 95% CI (1.13–2.21), *p* < 0.01]. Among motives for IPF, the perpetrator’s and/or the victim’s mental and/or physical illness significantly increased the perpetrators’ likelihood of suicide [OR = 2.75, 95% CI (1.85–4.10), *p* < 0.001].

Concerning the contextual level, 55.5% of perpetrators used a weapon as a firearm for IPFS, compared to 15.2% for IPF. The use of a gun emerged as a significant predictive risk factor IPFS [OR = 6.98, 95% CI (4.79–10.17), *p* < 0.001].

### 3.3. Hierarchical Logistic Regression Model of the IPFS’ Individual, Relational and Contextual Risk Factors

A hierarchical logistic regression analysis was conducted to establish the predictive role of some individual, relational, and contextual risk factors in the IPF perpetrators’ decision to commit suicide.

First, simple correlations were calculated between different risk factors and perpetrators’ suicide after IPF to analyze the relationship between variables. Regarding individual-level risk factors, the perpetrators’ suicide after IPF was found to be significantly associated with perpetrators’ age (r = 0.20, *p* < 0.01), being a member of law enforcement (r = 0.18, *p* < 0.01), and with suffering mental and/or physical illness (r = 0.10, *p* < 0.01). Interestingly, perpetrators suicide was negatively associated to substances use (r = −0.14, *p* < 0.01), to previous involvement in criminal behaviors (r = −0.16, *p* < 0.01) and to previous violent intimate relationships (r = −0.09, *p* < 0.05). At the relational level, perpetrators’ suicide emerged to be significantly associated to being older than the victims (r = 0.09, *p* < 0.05), having children in common (r = 0.11, *p* < 0.01), having children not in common (r = 0.09, *p* < 0.05) and with the presence of perpetrator and/or victim mental and/or physical illness (r = 0.20. *p* < 0.01). On the contrary, perpetrators’ jealousy (r = −0.17, *p* < 0.01) and quarrels and conflicts (r = −0.016, *p* < 0.01) were negatively associated to suicide after IPF. Regarding the contextual level risk factors, only using a firearm as a weapon to commit IPF was significantly associated with suicide (r = 0.42, *p* < 0.01).

All predictive factors that were significantly correlated to perpetrators’ suicide after IPF were entered into the hierarchical logistic regression (see Table 3).

Perpetrators’ individual-level risk factors, such as age, law enforcement membership, mental and/or physical illness, substances use, previous involvement in criminal behaviors, and violent relationships were simultaneously entered as covariates in Step 1. In Step 2, perpetrator-victim age gap, sons/daughters in or not in common, previous violence in the couple, quarrels and conflicts, jealousy, and perpetrator/victim mental and/or psychical illness were entered. Lastly, in Step 3 as a contextual level risk factor, the use of a firearm to commit IPF was entered to examine its unique effect on the perpetrators’ suicide.

In Step 1, only age (meaning being older) and law enforcement membership were statistically significant individual risk factors for perpetrators’ suicide after IPF. In the second step, relational level risk factors were entered, significantly increasing the variance explained, through only quarrels and conflicts and jealousy were negatively associated with the IPF perpetrators’ suicide. In the third step, using a firearm to commit IPF was entered, indicating that holding and using a gun to kill the victim, being older, and being a law enforcement member were significant risk factors that predict IPFS. Additionally, quarrels/conflicts and jealousy were significantly and negatively associated with the IPF perpetrators’ suicide. The full model explained 26.0% of the total variance of the IPF perpetrators’ suicide.

## 4. Discussion

Intimate partner femicide is a widespread phenomenon worldwide [69], representing the lethal and final act of prior gender-related violence [2], motivated by hatred, contempt, pleasure, or a sense of ownership of women, that leads to the light the differential fact of women’s violent death in a very poignant way, thus reframing it in terms of a special social and political problem [70].

In some cases, it could be followed by the perpetrator’s suicide [9,32]. However, few studies focused on the characteristics associated with the IPF perpetrators’ suicide. Most studies have mainly investigated risk factors for IPF to identify possible prevention and intervention strategies to protect women, often forgetting to analyze the characteristics of those IPF perpetrators that commit suicide afterward, a phenomenon known as IPFS.

To our knowledge, no studies analyzed and compared the IPF and IPFS perpetrators’ risk factors according to the ecological system theory [59,60,61]. So, we applied the ecological perspective to understand and investigate the possible role of the considered individual, relational, and contextual risk factors in the IPF perpetrators’ suicide.

To this aim, we carried out an archival study on IPF in Italy in the last decade, of 636 cases of IPF, 227 were IPFS (35.7%).

Following our first research question, the comparison between IPF and IPFS perpetrators’ risk factors highlighted, in line with prior studies, that using illicit drugs [6,11,12,13,20,44,48,49], having in the home a child who was not the killer’s biological child [20], previous crimes and violence within the couple [6,13,20,41,48,50], were all IPF characteristics; however, even though with a low frequency they also occurred in the IPFS cases.

Consistent with other studies on IPFS, the existence of a significant age gap between the perpetrator and the victim (7.04 years more for the perpetrator on average) [4,5,12,37] and being a member of law enforcement [46,47] were found to be significant risk factors for IPFS. Differently from the only two studies analyzing the role of the sons’/daughters’ presence [12,28], we found that having sons/daughters in common significantly increased the risk of IPFS; on the contrary, having sons/daughters who are not in common seems to be a predictive risk factor only for IPF.

Likewise, as underlined by a previous study [12], no prior history of criminal record or violent relationships and no history of drugs and alcohol use were significantly associated with the IPF perpetrators’ suicide.

Among motives for both IPF and IPFS, jealousy, quarrels and conflicts, and incapability to accept the end of the relationship were quite spread (even if they have not been found as risk factors for themselves) [3,4,6,9,10,12,35,37,52].

Furthermore, it would seem that only the presence of the perpetrator’ and/or the victim’s psychophysical illnesses [3,5,11,12,37,41,42] and the firearm possession [46,47] become a powerful and significant combination in which the probability to commit IPFS increases making the difference when compared with IPF or with other kinds of homicides in which the mental illness is rarely present [33].

About the second research question, the hierarchical logistic regression model results confirmed the perpetrators’ age role [4,6,12] and the law enforcement membership as significant individual risk factors for committing suicide after IPF; on the contrary, differently from other studies [3,6], our results did not highlight the possible role of the perpetrators’ mental and/or physical illness for IPFS. At the relational level, jealousy, quarrels, and violent behaviors in the couple were found as significant risk factors only for IPF, confirming the central role of the men’s possessiveness, and control over women as one of the main triggers in IPF cases [71]. Finally, at the contextual level, only the availability of a firearm to commit IPF seems to be significantly associated with IPFS. This result appears consistent with the literature on the phenomenon [3,4,5,6,9,10,11,35,38,39,40].

### Limitations

The present study has certain limitations. As in most archival research, information about IPF and IPFS was collected by searching and reading online newspaper articles, causing a high rate of missing cases as in the sample considered in the current study. Despite this, archival research using online newspaper articles is one of the methods adopted in this field [4,6,65], especially in countries where no available official data, reports, and statistics about the dimensions and risk factors were analyzed in the current study. Furthermore, to avoid a high number of missing information, each case of IPF/IPFS was monitored over time to have updated and more detailed information about the murder, the perpetrators’ characteristics, and the proceedings. Another possible limitation of the present research could rely on the difficulty in considering and including several individuals (e.g., social status, abuse and/or violence in childhood, witnessed IPV, parental conflicts and quarrels) relational (e.g., male partner economic control and/or dominance) and contextual dimensions (e.g., fear of stigmatization, norms that support IPV, the poor role of law, neighborhood violence), possibly justifying the relatively low percentage of variance explained by the hierarchical logistic regression model tested.

## 5. Conclusions

Causing the death of approximately two individuals, IPF and IPFS represent the most lethal forms of intimate partner abuse, leading to far-reaching effects on public health, families, friends, neighborhoods, and entire communities. This study contributes to the scant literature on the differences between IPF and IPFS fatal events and the need for prevention and intervention efforts. Consistent with other studies [11,12], our results suggest that IPFS involves perpetrators with a different individual, relational, and contextual risk factors than the IPF perpetrators. While risk factors associated with IPF are pretty much known, further research should consider the role of other possible risk factors to explain why only some authors commit both IPF and suicide.

Our results summarize the suicidal killer profile as an ordinary man without a history of criminal records or violent relationships, nor a previous history of drug or alcohol use [12]. However, in our study, being a law enforcement member at the individual level and firearm availability at the contextual level are significant risk factors for committing suicide after IPF. We could hypothesize that consistent with previous studies [46,47], law enforcement members are more at risk of committing homicide–suicide in domestic violence than their civilian counterparts due to several job-related risk factors.

If on one side, our results evidenced a suicidal killer as an ordinary man, and on the other side, they seem to suggest the need to better consider and investigate, consistently with Dawson [3], the possible role of the perpetrators’ psychological wellbeing and psychophysical status in the decision to commit suicide after IPF.

Our findings underline that law enforcement members and those who possess a firearm are more likely to commit suicide after the IPF.

Regarding IPFS prevention, it could be crucial to focus on the employees’ psychological health, especially those equipped with a firearm. In fact, despite few studies, law enforcement members showed the presence of some job-related risk factors [47] that needed to be assessed and monitored over time.

Considering this, it could be helpful to develop, implement and evaluate the effectiveness of Behavioral Health Training functional to intercept signs, symptoms, and risk factors of mental health issues [46]. Because of IPFS’s devastating effects on families and communities [6,15], it is advisable to emphasize the need for the government, healthcare system, and community interventions to raise public awareness and identify, recognize, and manage significant risk factors for IPFS. In this sense, it could be helpful to work on the development of self-assessment actuarial tools to detect possible “alarm bells” for IPFS and to monitor the psychological well-being of the people and of those who possess the firearm.

Furthermore, further studies are needed considering the scant literature on the topic and the few variables considered to explain the phenomenon. Future studies should identify the critical risk factors related to the IPF perpetrator’s suicide and investigate which individual, relational and contextual dimensions operate as protective factors for the IPF perpetrators, by adopting quantitative and qualitative methods.

For example, future research could focus on the contributing role of some cognitive and emotional dimensions associated with IPF perpetrators’ choice to commit suicide. IPFS perpetrators should have more to lose and be characterized by high levels of apprehension, fear of social disapproval, social margination, stigmatization of the consequences, and remorse related to killing the victim [3,4,9,72,73]. These cognitive and emotional factors, however, need more investigation. Further studies also should maybe focus on the possible mediating and/or moderating role of the perpetrator’s social status [9,12], as it could be, a critical factor in perpetrators’ suicide after IPF.

Finally, future research should be designed to collect multiple data sources [65], for example integrating data from newspaper articles with structured or semi-structured interviews with family, friends, and witnesses or also applying the techniques of the psychological autopsy.

## Figures and Tables

**Table 1 ijerph-19-10431-t001:** Descriptive statistics of risk factors in IPF.

**Individual level risk factors**			
Perpetrator age		M = 51.9 (SD = 17.7)	
IPFS			35.7
Perpetrator nationality	Italian		81.4
	Foreign		
Occupation	Unemployed/retired		35.2
	Low specialization		50.0
	High specialization		14.8
Previous crimes	Yes		17.6
	No		82.4
Substance use	Yes		12.6
	No		82.4
Physical or mental disorders	Yes		8.8
	No		91.2
Previous violent relationships	Yes		3.5
	No		96.5
**Relationship level risk factors**			
Age difference		M = 7.04 (SD = 5.49)	
Type of relationship	Husband/cohabitant		65.7
	Ex-husband/ex-cohabitant		18.6
	Boyfriend		5.8
	Ex-boyfriend		9.9
Children in common	Yes		57.7
	No		42.3
Children not in common	Yes		21.5
	No		78.5
Motives of IPF/IPFS	Jealousy		19.5
	Incapability to accept the end of the relationship		35.7
	Quarrels and conflicts		14.6
	Perpetrator/victim’s mental and physical illness		1.6
Previous violence in the couple	Yes		38.5
	No		61.5
**Contextual level risk factors**			
Weapon used for IPF	Yes		29.6
	No		70.4
Victim’s previous help request	Yes		25.3
	No		74.7

**Table 2 ijerph-19-10431-t002:** Differences in individual, relational, and contextual risk factors between IPF and IPFS perpetrators.

	Risk Factors		IPFS		
			Yes (N = 227)	No (N = 409)	
Individual level					
	Age		M = 56.6 (SD = 17.9)	M = 49.3 SD = 17.0	25.40 ***
	Occupation				OR (C.I.)
		Unemployed/retired	43.3	38.1	1.24 (0.87–1.77)
		Low specialization	39.0	45.7	0.82 (0.58–1.18)
		High specialization	17.7	16.3	1.13 (0.69–1.85)
	Law enforcement		16.7	5.9	3.23 *** (1.88–5.34)
	Previous crimes		9.7	22.0	0.38 *** (0.23–0.63)
	Substance use		6.2	16.1	0.34 *** (0.19–0.62)
	Mental and/or physical illness		30.4	13.7	2.75 *** (1.85–4.10)
	Previous violent relationships		1.3	4.6	0.28 * (0.08–0.94)
Relationship level					
	Perpetrator/victim age gap		85.5	77.9	1.68 * (1.07–2.63)
	Children in common		64.8	53.8	1.58 ** (1.13–2.21)
	Children not in common		16.7	24.2	0.63 * (0.42–0.95)
	Motives of IPF/IPFS	Jealousy	10.6	24.4	0.37 *** (0.23–0.59)
		Incapability to accept the end of the relationship	38.8	34.0	1.23 (0.88–1.72)
		Quarrels and conflicts	7.0	18.8	0.33 *** (0.19–0.58)
		Perpetrator/victim’s mental and physical illness	30.4	13.7	2.75 *** (1.85–4.10)
	Previous violence in the couple		28.2	44.3	0.51 *** (0.36–0.72)
Contextual level					
	Weapon used for IPF		55.5	15.2	6.98 *** (4.79–10.17)
	Previous help requests by the victim		21.6	27.4	0.73 (0.50–1.07)

Notes: * *p* < 0.05, ** *p* < 0.01, *** *p* < 0.001. OR = Odd Ratio, CI = Confidence Interval.

**Table 3 ijerph-19-10431-t003:** Hierarchical regression including individual, relational, and contextual risk factors for IPFS.

	Step 1			Step 2			Step 3		
	B	SE B	β	B	SE B	β	B	SE B	β
Perpetrator age	0.01	0.00	0.18 ***	0.00	0.00	0.13 **	0.00	0.00	0.10 *
Law enforcement member	0.31	0.07	0.19 ***	0.30	0.06	0.19 ***	0.17	0.06	0.11 *
Perpetrator mental/physical illness	0.10	0.07	0.06	−0.01	0.09	−0.00	0.02	0.08	0.01
Substances use	−0.08	0.06	−0.06	−0.04	0.06	−0.03	−0.02	0.06	−0.01
Previous criminal behaviors	−0.08	0.05	−0.07	−0.05	0.05	−0.04	−0.04	0.05	−0.03
Previous violent relationships	−0.13	0.11	−0.05	−0.06	0.11	−0.02	−0.05	0.09	−0.02
Perpetrator/victim age gap				0.04	0.05	0.03	0.02	0.05	0.01
Children in common				0.02	0.04	0.02	0.01	0.04	0.01
Children not in common				−0.04	0.05	−0.03	−0.06	0.05	−0.05
Previous violence in the couple				−0.06	0.04	−0.06	−0.07	0.04	−0.07
Quarrels and conflicts				−0.23	0.06	−0.17 ***	−0.16	0.05	−0.12 **
Jealousy				−0.17	0.05	−0.14 ***	−0.14	0.05	−0.12 **
Perpetrator/victim mental or physical illness				0.04	0.08	0.04	0.03	0.07	0.03
Firearm							0.38	0.04	0.36 ***
R^2^	0.10			0.14			0.26		
F	10.96 ***			7.57 ***			14.52 ***		
ΔR^2^	0.09			0.13			0.24		

Notes: CI = confidence interval; LL = lower limit; UL = upper limit. The nominal significance level (i.e., *p* < 0.05) was corrected according to the Bonferroni procedure and set at *p* < 0.008 for Step 1, to *p* < 0.007 for Step 2 and *p* < 0.05 for Step 3. * *p* < 0.0083. ** *p* < 0.0071. *** *p* < 0.05.

## Data Availability

The data presented in this study are accessible only by directly contacting the authors. Due to anonymity, ethical, and personal reasons, the data are not publicly available.

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
