# Peer review of "Risk Factors for Intimate Partner Femicide–Suicide in Italy: An Ecological Approach"

_ijerph, 2022, doi:10.3390/ijerph191610431_

Round 1

Reviewer 1 Report

The study examined intimate partner femicide and intimate partner femicide/suicide. This was accomplished through the use of archival data (newspapers, and cases). The results ultimately highlighted risk factors associated with the perpetrators' decision. The authors then highlighted how important these findings were for treatment, rehabilitation, etc. The literature review is thorough, the methods are clear, and it is well written. 

The study overall is strong, my only critique is that the authors' don't really explain is justify why the data they are using is the best, and/or most suitable data source for this study. They just say this is the data we are using, why this data over other(s)? If there aren't any others, just state that clearly. 

Author Response

We thank the reviewer for his/her request for further details. We inserted a justification related to our statement (lines 168-170), as archival data were the most suitable data source for our study since, as underlined in the study limitations section, no official data, reports, or statistics are available in Italy. Even if some entities (e.g., ISTAT and EURES) collect and report data on female homicide prevalence, as far as we know, none of them included all the information and the variables collected and analyzed for the present study.

Reviewer 2 Report

This study deals with an important and under-researched topic and can therefore be seen as a significant contribution to current state of research. I have the following comments/suggestions for improvement:

Literature review:

-        As the authors mention that no study so far has compared IPF and IPFS they might have missed the study of Liem and Roberts (2009), DOI: 10.1177/1088767909347988.  Furthermore, the detailed description of the studies from Ghana and Jamaica seems somewhat disconcerting, as these countries culturally differ from Western European ones. It might be better to give preference to European studies and mention the others only in passing. Finally, more information should be given on the ecological approach, as it even appears in the title of the study.

Data:

The study draws on an extensive and impressive database collected through media reports. Although the information was supplemented by NGO and other information, this data is a kind of a weak point of the whole study. As the authors themselves point out in the limitations section, there were several missing values that carry a risk of bias. I would also argue that information given in newspapers is not per see the correct information but might be speculative. The authors argue that "archival research using online newspaper articles is one of the most commonly used methods in the field [3, 5, 60], especially in countries where official data, reports and statistics on the dimensions and risk factors analysed in the current study [are not available]". There are several Western European countries such as the Netherlands and the Scandinavian countries that have extensive databases. Moreover, even in countries without regular data collection, data can be formed on the basis of forensic, police and court records, as has been done in Switzerland in the framework of the Swiss Homicide Monitor. Why is such data collection not possible in Italy?  

Analysis:

It might be possible that the inclusion of older couples has an influence on the results in terms of "mercy killings". 

It seems kind of strange that more than 30% of IPFS perpetrators had mental and/or physical illness but that this item is not significant in regression analyses. Is there any explanation for this? Might it have to do with the rather broad definition of the question (no information is given on what falls under this term)? Separating physical and mental illness into two separate variables might lead to a different result.

Finally, I wonder if there is an intervening effect between gun and law enforcement affiliation. This possible influence could easily be tested by including a corresponding variable (gun*law enforcement).

Author Response

Literature review:

Q1: As the authors mention that no study so far has compared IPF and IPFS they might have missed the study of Liem and Roberts (2009), DOI: 10.1177/1088767909347988. 

R: We thank the reviewer for his/her suggestion. We added the missing study accordingly.

Q2: Furthermore, the detailed description of the studies from Ghana and Jamaica seems somewhat disconcerting, as these countries culturally differ from Western European ones. It might be better to give preference to European studies and mention the others only in passing. 

R: We thank the reviewer for his/her suggestion. As stated in the introduction, we focused on studies investigating the characteristics of IPF and IPFS perpetrators according to a gender-sensitive perspective as described by other authors (Adinkrah, 2014; Dayan, 2018; Dawson, 2005) and distinguishing IPF/IPFS from Homicide-Suicide. As underlined by Standish & Weil (2021) and as far as we know, at the European and international level, almost all studies included IPF and IPFS in homicide and homicide/suicide phenomena, thus denying or underestimating the IPF and IPFS phenomena’s strong gendered nature.

Q3: Finally, more information should be given on the ecological approach, as it even appears in the title of the study.

R: We added an explanation (lines 150-155) about our rationale for choosing the ecological approach as a theoretical framework to explore how the different individual, relational, and contextual risk factors should differentiate perpetrators who commit IPF from those who commit suicide after IPF.

Data:

Q4: The study draws on an extensive and impressive database collected through media reports. Although the information was supplemented by NGO and other information, this data is a kind of a weak point of the whole study. As the authors themselves point out in the limitations section, there were several missing values that carry a risk of bias. I would also argue that information given in newspapers is not per see the correct information but might be speculative. The authors argue that "archival research using online newspaper articles is one of the most commonly used methods in the field [3, 5, 60], especially in countries where official data, reports and statistics on the dimensions and risk factors analysed in the current study [are not available]". There are several Western European countries such as the Netherlands and the Scandinavian countries that have extensive databases. Moreover, even in countries without regular data collection, data can be formed on the basis of forensic, police and court records, as has been done in Switzerland in the framework of the Swiss Homicide Monitor. Why is such data collection not possible in Italy?  

R: We thank the reviewer for his/her question. Unfortunately, as underlined in the study limitations section, no official data, reports, or statistics are available in Italy. Even if some entities (e.g., ISTAT and EURES) collect and report data on female homicide prevalence, as far as we know, none of them included all the information and the variables collected and analyzed for the present study. As IPV and IPF/IPFS are spread phenomena involving an increasing number of women victims, we hope that our research could shed light on the need to institute an official national observatory for femicide.

Analysis:

Q5: It might be possible that the inclusion of older couples has an influence on the results in terms of "mercy killings". It seems kind of strange that more than 30% of IPFS perpetrators had mental and/or physical illness but that this item is not significant in regression analyses. Is there any explanation for this? Might it have to do with the rather broad definition of the question (no information is given on what falls under this term)? Separating physical and mental illness into two separate variables might lead to a different result.

R: We thank the reviewer for this interesting suggestion. Our database covers nineteen years. Unfortunately, pieces of information about perpetrators and victims' types of illness were not always available also after checking for various data sources (local, regional and national online media and newspapers NGO’s report, the27ora website) and monitoring each IPF/IPFS case over time to detect any possible additional information.  Besides this, we included in our sample all cases of IPF and IPFS that occurred in Italy regardless of perpetrators’ age, as we aimed to investigate the role of some individual, relational, and contextual risk factors in IPF perpetrators’ decision to commit suicide after femicide. Furthermore, even if consistent with the reviewer, it is conceivable that risk factors patterns may differ across IPF/IPFS perpetrators’ age groups, we included older IPF and IPFS perpetrators in our analyses as we considered the so-called phenomenon of ‘mercy killings’ as one of the possible motives for IPFS,  as also underlined by Salari and Sellitto (2016), all IPFS cases as the lethal form of intimate partner violence. Consistent with the reviewer, we believe that in future studies, it would be crucial to have more accurate data about IPF and IPFS perpetrators’ physical and/or mental illness by interviewing their family, friends, colleagues, and health professionals they eventually had contact. We believe that testing the separate role of physical and mental illness across IPF and IPFS perpetrators should have a crucial preventive value and help shed light on these variables' eventual contributing role in both IPF and IPFS.

Q6: Finally, I wonder if there is an intervening effect between gun and law enforcement affiliation. This possible influence could easily be tested by including a corresponding variable (gun*law enforcement).

R: We thank the reviewer for his/her suggestion. We tested the possible interaction between gun*law enforcement membership by including the variable in our hierarchical regression model. The result showed that gun*law enforcement membership was not a significant predictor of IPFS (β= -.018, p=.765). Furthermore, gun*law enforcement membership did not contribute to the variance of IPFS explained, and when entered in the fourth step, results remained unchanged.

Reviewer 3 Report

The articles states its main objective clearly and the results and very well stated as well as the methodology. 

At an theoretical leve, with influnece in the discussion of results, I believe that the article completely disregards the structural elements of IPF and IPFS which is gender-based. Even when the aspects like jealousy and domination are stated, the gender-based component is completely disregarded. I strongly recommend that the article should introduce articulations, when possible and pertinent, between IPF and IPFS and gender-based violence which is something well established within the scientific community. Even the use of the terminology femicide, presupposes the gender element which in the articles is disregarded.

Author Response

We thank the reviewer for his/her suggestion. We accordingly have pointed out the gender-based nature of IPF and IPFS both in the introduction and in the discussion section.